# Psychological Determinants of COVID-19 Vaccine Acceptance among Healthcare Workers in Kuwait: A Cross-Sectional Study Using the 5C and Vaccine Conspiracy Beliefs Scales

**DOI:** 10.3390/vaccines9070701

**Published:** 2021-06-25

**Authors:** Mariam Al-Sanafi, Malik Sallam

**Affiliations:** 1Department of Clinical Pharmacy, Farwaniya Hospital, Kuwait City 25210, Kuwait; mb.alsanafi@paaet.edu.kw; 2Department of Pharmaceutical Sciences, Public Authority for Applied Education and Training, College of Health Sciences, Safat 13092, Kuwait; 3Department of Pathology, Microbiology and Forensic Medicine, School of Medicine, the University of Jordan, Amman 11942, Jordan; 4Department of Clinical Laboratories and Forensic Medicine, Jordan University Hospital, Amman 11942, Jordan; 5Department of Translational Medicine, Faculty of Medicine, Lund University, 22184 Malmö, Sweden

**Keywords:** vaccine hesitancy, anti-vaccine, intention to vaccinate, vaccine rejection, SARS-CoV-2 prevention

## Abstract

Acceptance of coronavirus disease 2019 (COVID-19) vaccination appears as a decisive factor necessary to control the ongoing pandemic. Healthcare workers (HCWs) are among the highest risk groups for infection. The current study aimed to evaluate COVID-19 vaccine acceptance among HCWs in Kuwait, with identification of the psychological determinants of COVID-19 vaccine hesitancy. The study was conducted using an online anonymous survey distributed between 18 March 2021 and 29 March 2021. The sampling strategy was convenience-based depending on chain-referral sampling. Psychological determinants of COVID-19 vaccine acceptance were assessed using the 5C subscales and the Vaccine Conspiracy Beliefs Scale (VCBS). The total number of study participants was 1019, with the largest group being physicians (28.7%), pharmacists (20.2%), dentists (16.7%), and nurses (12.5%). The overall rate for COVID-19 vaccine acceptance was 83.3%, with 9.0% who were not willing to accept vaccination and 7.7% who were unsure. The highest rate for COVID-19 vaccine acceptance was seen among dentists (91.2%) and physicians (90.4%), while the lowest rate was seen among nurses (70.1%; *p* < 0.001). A higher level of COVID-19 vaccine hesitancy was found among females, participants with a lower educational level, and HCWs in the private sector. A preference for mRNA vaccine technology and Pfizer-BioNTech COVID-19 vaccine was found among the majority of participants (62.6% and 69.7%, respectively). COVID-19 vaccine hesitancy was significantly linked to the embrace of vaccine conspiracy beliefs. The highest 5C psychological predictors of COVID-19 vaccine acceptance were high levels of collective responsibility and confidence, and lower levels of constraints and calculation. The VCBS and 5C subscales (except the calculation subscale) showed acceptable levels of predicting COVID-19 vaccine acceptance based on receiver operating characteristic analyses. The participants who depended on social media platforms, TV programs, and news releases as their main sources of knowledge about COVID-19 vaccines showed higher rates of COVID-19 vaccine hesitancy. An overall satisfactory level of COVID-19 vaccine acceptance was seen among HCWs in Kuwait, which was among the highest rates reported globally. However; higher levels of vaccine hesitancy were observed among certain groups (females, nurses and laboratory workers, HCWs in the private sector), which should be targeted with more focused awareness programs. HCWs in Kuwait can play a central role in educating their patients and the general public about the benefits of COVID-19 vaccination to halt the spread of SARS-CoV-2, considering the high rates of vaccine hesitancy observed among the general public in Kuwait and the Middle East.

## 1. Introduction

Vaccine hesitancy is considered among the top public health threats globally [1,2]. In the context of coronavirus disease 2019 (COVID-19) prevention, vaccine hesitancy can be a major challenge facing the efforts needed for proper control of the pandemic [3,4,5].

Successful vaccination campaigns appear to be one of the most important public health measures to curb the rise in cases with its associated burden on healthcare systems [6,7]. However, the rejection and hesitancy toward COVID-19 vaccination can be a major barrier in the prevention efforts aiming to alleviate the devastating health, economic, and psychological consequences of this unprecedented pandemic [8,9,10].

Several COVID-19 vaccines with remarkable safety and efficacy profiles have been approved for emergency use to prevent severe acute respiratory syndrome coronavirus 2 (SARS-CoV-2) acquisition, spread, and to halt the occurrence of COVID-19 severe cases, hospitalization, and mortality [11].

These vaccines can be divided based on type (technology) and developers (manufacturers) into (1) messenger RNA (mRNA) vaccines (e.g., Pfizer-BioNTech COVID-19 vaccine (tozinameran), and Moderna COVID-19 vaccine); (2) adenovirus vector vaccines (e.g., Oxford-AstraZeneca COVID-19 vaccine (Covishield), Sputnik V COVID-19 vaccine, and Johnson & Johnson COVID-19 vaccine); (3) inactivated virus vaccines (e.g., Sinopharm COVID-19 vaccine); (4) subunit vaccines (e.g., EpiVacCorona) [12,13,14,15,16,17].

Vaccine hesitancy can be described as a complex phenomenon that varies based on place, time, and type of vaccines [18]. This complexity led to challenges and ambiguities in the scientific efforts that aimed to establish a consensus on its definition [19].

For example, Eve Dubé et al. defined vaccine-hesitant individuals as the heterogeneous group in the middle of the attitude spectrum toward vaccination, with active demand for vaccines to complete refusal of all vaccines at both ends of the spectrum. Thus, vaccine-hesitancy entails the refusal of some vaccines but agreeing to take others based on the aforementioned definition. It also entails delaying or accepting vaccines but being unsure in doing so [19]. Another definition conceived by Patrick Peretti-Watel et al. defined vaccine hesitancy as “a decision-making process that depends on people’s level of commitment to healthism/risk culture and on their level of confidence toward health authorities and mainstream medicine” [20].

Herein, we opted to use the term COVID-19 vaccine hesitancy to describe the individuals who displayed reluctance (uncertain intention manifested as “maybe” as an answer) or refusal to receive COVID-19 vaccines (clear intention no to receive the vaccine), despite the availability of vaccination services [18].

The psychological factors appear decisive in the individual attitude toward vaccination, in general, and toward COVID-19 vaccination, in particular [21,22,23,24]. This is mainly related to the psychological impact of the current pandemic, which was accompanied by an infodemic. COVID-19 infodemic involved the circulation of misinformation and disinformation involving many aspects of the disease including vaccination [25,26,27].

Deciphering the psychological factors that can drive the individual attitude toward COVID-19 vaccination is invaluable to designing evidence-based strategies that would help to combat vaccine hesitancy [28]. Analyzing the psychological factors can be facilitated by the use of validated and robust measures of the subject attitude toward vaccination [29]. Two scales have been suggested, which include the 5C scale to evaluate the psychological antecedents of vaccination and the vaccine conspiracy beliefs scale (VCBS), which can assess the role of conspiracy beliefs in rejection of vaccination [30,31,32]. The 5C scale in particular has been shown to be superior, compared to other scales, in order to explain the variance of self-reported vaccination behavior [33].

In brief, the 5C scale aims to assess the following determinants: (1) confidence in safety and efficacy of vaccines and the trust in the providers including policymakers and the healthcare workers providing the service; (2) complacency, which is defined by the low-perception of disease risk; (3) constraints, which include the physical and psychological barriers rendering vaccination inconvenient; (4) calculation, which entails active engagement in searching for information about the vaccine and its utility; (5) collective responsibility, defined by the extent of willingness to benefit others by receiving vaccination to help in achieving herd immunity [31].

In addition, studying the extent of embracing vaccine conspiracy beliefs appears to be of high value in some regions (e.g., the Middle East). This value of such studies can be justified by the high prevalence of misinformation and beliefs in conspiracies regarding several aspects of COVID-19 including vaccination in this region [23,34]. 

The study of COVID-19 vaccine hesitancy among healthcare workers (HCWs) is invaluable for the following reasons: HCWs represent a group of individuals who are exposed to a higher risk of infections including COVID-19 and a more hazardous environment [35]. In addition, HCWs can be viewed as a trusted group that should provide vaccine-related information to the patients and the general public as well [36].

The Middle East represents a region with the lowest rates of COVID-19 vaccine acceptance worldwide [27,37]. Kuwait is a Middle Eastern country with a population of more than 4,200,000 people, of which about 65% are of non-Kuwaiti citizenships as of 2019 [38]. In Kuwait, the total number of active COVID-19 cases reached more than 14,000 cases by the end of April 2021. In the country, vaccination against COVID-19 started in December 2020, and the total number of vaccine doses given by the end of April 2021 was 1.3 million, with about 38,000 fully vaccinated individuals [39]. Two vaccines were approved in the country: Pfizer-BioNTech COVID-19 vaccine and Oxford-AstraZeneca COVID-19 vaccine [40].

In this study, the major aims were as follows: (1) to assess the overall acceptance rates of COVID-19 vaccines among HCWs in Kuwait; (2) to evaluate the potential differences in attitude toward COVID-19 vaccines among different variables including occupation (doctors, nurses, pharmacists, etc.); (3) to assess the psychological antecedents of COVID-19 vaccination and the effect of embracing COVID-19 vaccine conspiracy beliefs in vaccine acceptance among HCWs in the country. Other minor objectives included assessing the roles of sources of information, vaccine types, and vaccine manufacturers in preferences and attitudes toward COVID-19 vaccination. 

## 2. Materials and Methods

### 2.1. Study Design

A cross-sectional survey was conducted to analyze COVID-19 vaccine acceptance rates and the potential psychological factors of COVID-19 vaccine hesitancy among HCWs in Kuwait. The survey was conducted between 18 March 2021 to 29 March 2021. The convenience sampling strategy was utilized based on chain-referral sampling, starting with contacts of the first author and sharing the survey on Facebook, Twitter, LinkedIn, etc., besides the free messaging service WhatsApp.

Based on the estimate of about 60,000 HCWs in Kuwait, and considering a margin of error = 4.0% (with 95% confidence interval), the calculated minimum sample size was 595 participants [41,42].

The questionnaire items were adopted from previous studies tackling the same objectives [27,31,43]. The study questionnaire items were offered both in Arabic and English languages. The study questionnaire’s exact wording is available in (Appendix A). 

### 2.2. Overview of the Questionnaire Items

The questionnaire comprised an introductory section with consent to participate. This was followed by 31 mandatory items divided as follows: a section on the sociodemographic data of the participants (age, sex, occupation (physician, dentist, pharmacist, nurse, laboratory technician, other), educational level (under- or post-graduate), workplace (public vs. private), history of chronic, and a previous COVID-19 diagnosis of the participant or a family member); an item on the participant intent for COVID-19 vaccination (Did you receive a COVID-19 vaccine/Do you intend to receive a COVID-19 vaccine?); an item assessing complete opposition to vaccination in general; an item assessing the belief in a conspiracy about SARS-CoV-2 origin; a section on the preference for COVID-19 vaccine based on technology, and the preference for COVID-19 vaccine based on developer; an item assessing the single main source of information about COVID-19 vaccination.

For the 5C scale, a section that comprised 10 items was included (involving 2 items for each determinant). These items were selected based on a previous protocol by Cornelia Betsch et al. [31]. Confidence was assessed using the following items: (1) regarding vaccines, I am confident that public authorities decide in the best interest of the community; (2) vaccinations are effective. Complacency was assessed using the following items: (1) COVID-19 is not so severe that I should be vaccinated; (2) my immune system is so strong, it also protects me against it. Constraints were assessed using the following items: (1) for me, it is inconvenient to be vaccinated; (2) visiting the doctor makes me feel uncomfortable; this keeps me from being vaccinated. The calculation was assessed using the following items: (1) for each and every vaccination, I closely consider whether it is useful for me; (2) it is important for me to fully understand the topic of vaccination before I receive vaccination. Collective responsibility was assessed using the following items: (1) I receive vaccination because I can also protect people with a weaker immune system; (2) vaccination is a collective action to prevent the spread of diseases. The internal consistency of the 5C subscales was checked through Cronbach’s alpha values. 

The final section comprised seven items used to measure the Vaccine Conspiracy Beliefs Scale (VCBS), which was minimally modified to target COVID-19 vaccination [27,32,43].

For both the 5C scale and VCBS items, 7-point Likert scale was used with “strongly disagree” response given a minimum score of 1, “neutral” response given a score of 4, and the maximum score of 7 given to “strongly agree” response. The exceptions were confidence and collective responsibility, for which the scores were reversed.

### 2.3. Major Measures in the Study

#### 2.3.1. Willingness to Accept COVID-19 Vaccination

The major outcome measure in this study was the intent for COVID-19 vaccination, with three possible responses (yes vs. no vs. maybe). Vaccine hesitancy in relation to the 5C psychological antecedents was assessed by dividing the study population into two groups: participants with a clear intent to receive the vaccine (Yes response) and participants who were hesitant or rejected COVID-19 vaccination (Maybe or No responses). 

#### 2.3.2. The Role of Psychological Antecedents and Conspiracy Beliefs in COVID-19 Vaccine Rejection/Hesitancy

The 5C scale was used to assess the five psychological antecedents for vaccination using five subscales, each of which comprised two items: confidence subscale; complacency subscale; constraints subscale; calculation subscale; and collective responsibility subscale. The five subscales were assessed for internal consistency with the following Cronbach’s alpha values of 0.676 for the confidence subscale, 0.671 for the complacency subscale, 0.657 for the constraints subscale, 0.464 for the calculation subscale, and 0.843 for the collective responsibility subscale. The role of embracing vaccine conspiracy beliefs was assessed using VCBS that showed a Cronbach’s alpha value of 0.937. The correlation between COVID-19 vaccine acceptance with the 5C subscales was evaluated using multinomial logistic regression with the intention for COVID-19 vaccination as the dependent variable, each of the 5C subscales as the fixed factors, and the following as covariates: age (<34 years vs. ≥34 years); sex; nationality; occupation; workplace; and educational level. 

#### 2.3.3. Dichotomous Classification of the 5C Subscales for Multinomial Regression Analysis

Each of the 5C subscales and the VCBS was divided into two categories based on the mean value as follows: confidence, complacency, and constraints subscales (<5.0 vs. ≥5.0); calculation subscale (<11.0 vs. ≥11.0); collective responsibility subscale (<4.0 vs. ≥4.0); VCBS (<19.0 vs. ≥19.0).

#### 2.3.4. COVID-19 Vaccine Hesitancy in Relation to VCBS

The correlation between the intention to receive COVID-19 vaccination and vaccine conspiracy beliefs was evaluated using univariate analysis with the intention for COVID-19 vaccination as the dependent variable, VCBS as the fixed factor, and the following as covariates: age; sex; nationality; occupation; educational level; workplace.

### 2.4. Ethical Permission

This study was approved by the Ministry of Health in Kuwait—Assistant undersecretary for planning and quality (Reference Number: 2264; Research Number: 2021/1668). The study was also approved by the general director of Farwaniya Hospital in Kuwait City. The informed consent was obtained by the mandatory item at the beginning of the questionnaire asking for the agreement from the participant to be part of the study. Participation in the study was anonymous, and all collected data were treated with full privacy and confidentiality. 

### 2.5. Statistical Analysis

All statistical analyses were conducted using IBM SPSS Statistics for Windows, Version 22.0. Armonk, NY, USA: IBM Corp. To test the possible associations between categorical variables, we used the chi-squared test (χ^2^). To test the possible associations between scale variables (age, VCBS) and categorical variables, we used a two-tailed Mann–Whitney *U* test, the Kruskal–Wallis (K–W) test, and the one-way analysis of variance (ANOVA) as appropriate. Linear-by-linear test for association (LBL) was used to estimate the change in COVID-19 vaccine preference over the survey time. Univariate and multinomial regression analyses were used as appropriate. The cutoff point for statistical significance was determined at *p* < 0.050.

To evaluate the ability of 5C subscales in the prediction of COVID-19 vaccine hesitancy, we used the receiver operator characteristic (ROC) curve with the calculation of the area under the curve (AUC).

## 3. Results

### 3.1. Characteristics of the Study Participants

The total number of study participants that were included in the final analysis was 1019 HCWs. The overall characteristics of the study participants are illustrated in (Table 1). The mean age for the whole study population was 34 years (median: 32 years, standard deviation: 9.7, interquartile range: 26–39 years). The majority of participants were females, of Kuwaiti nationality and working in the public healthcare sector. Non-Kuwaiti participants represented 23 countries, with the following countries having the highest number of participants: Egypt (*n* = 45), India (*n* = 40), Syria (*n* = 27), and Philippines (*n* = 25). Additionally, 33 participants reported unknown/stateless responses for nationality.

The study participant characteristics varied according to occupation, with the most notable difference as follows: in terms of age, dentists were the youngest group, while physicians were the oldest; in terms of sex, males formed the majority among dentists, the distribution was almost equal among physicians, while females represented the majority among nurses, pharmacists, laboratory technicians, and the “other” group. The Kuwaiti nationality predominated across all occupations except for nurses that had a majority of non-Kuwaiti nationality, and postgraduate education predominated among physicians (Table 2).

### 3.2. An Overall High Rate for Acceptance of COVID-19 Vaccination among HCWs in Kuwait

The overall intent to receive COVID-19 vaccination among the study participants was 83.3% (*n* = 849, who answered yes), compared to 9.0% who rejected COVID-19 vaccination (*n* = 92, who answered no) and 7.7% (*n* = 78, who answered maybe).

Stratified by different characteristics, a higher acceptance of COVID-19 vaccination was seen among males, participants of Kuwaiti nationality, physicians and dentists, participants with postgraduate educational level, and participants working in the public healthcare sector (Table 3). Age was not significantly associated with COVID-19 vaccine acceptance among the study participants (*p* = 0.208; K-W).

### 3.3. A Majority of HCWs in Kuwait Preferred Messenger RNA-Based and Pfizer-BioNTech Vaccines

Based on vaccine class, the majority of the study participants preferred mRNA-based vaccines (*n* = 638, 62.6%), followed by inactivated vaccines (*n* = 253, 24.8%). Males had a higher preference for mRNA vaccines, compared to females, while physicians and dentists had a higher preference for mRNA vaccines, compared to nurses, who had the highest preference for inactivated vaccines (Table 4). Based on vaccine developer, the majority of the study participants preferred the Pfizer-BioNTech vaccine (*n* = 710, 69.7%), followed by the Oxford-AstraZeneca vaccine (*n* = 193, 18.9%, Table 5).

The intention to receive COVID-19 vaccination (yes as the answer) was associated with a higher likelihood of preference toward mRNA-based vaccines (69.7%) and toward preferring Pfizer-BioNTech vaccine (74.8%). Conversely, COVID-19 vaccine hesitancy showed a lower preference to such vaccines, where 21.7% of those who answered no and only 33.3% of those who answered maybe preferred mRNA vaccines, and only 38.0% who answered no and 51.3% of those who answered maybe preferred Pfizer-BioNTech vaccine (*p* < 0.001 for the two comparisons among the three groups; χ^2^ test, Figure 1).

By dividing the survey period into two halves (18 February 2021 to 23 March 2021 vs. 24 March 2021 to 29 March 2021), we found a significant drop in preference toward Oxford-AstraZeneca vaccine, in comparison to the Pfizer-BioNTech vaccine (23.3% in the first half to 17.5% in the second half; *p* = 0.047, LBL). A significant difference was also found upon conducting the same comparison by each day of the study survey (*p* = 0.041; LBL, Figure 2).

### 3.4. The Conspiratorial Belief Regarding SARS-CoV-2 Origin and Anti-Vaccination Altogether Were Associated with COVID-19 Vaccine Hesitancy

Overall, less than a third of the whole study population believed that SARS-CoV-2 had a human-made origin (*n* = 300, 29.4%), compared to 46.8% who believed in the natural origin of the virus (*n* = 477), and 23.7% who reported no opinion (*n* = 242).

The belief that the virus had a human-made origin was significantly associated with less intention to receive COVID-19 vaccination (67.3%), compared to 91.6% among those who believed in the natural origin of the virus, and 86.8% among those who did not have an opinion regarding the origin of the virus (*p* < 0.001; χ^2^ test, Figure 3).

For assessment of the prevalence of anti-vaccination altogether among the whole study population, we found that only 4.1% were anti-vaccination (*n* = 42), compared to 5.3% who did not have an opinion (*n* = 54), and 90.6% who were against anti-vaccination (*n* = 923).

Among the participants who were anti-vaccination, only 14.3% expressed an intent to receive COVID-19 vaccination (*n* = 6), compared to 50.0% in the “no opinion group” (*n* = 27), and 88.4% among those who rejected anti-vaccination (*n* = 816, *p* < 0.001; χ^2^ test, Figure 3).

### 3.5. COVID-19 Vaccine Conspiracy Beliefs Were Associated with a Significantly Lower Intent to Receive COVID-19 Vaccination among HCWs in Kuwait

Based on the assessment of the VCBS, the highest mean score was found among the participants who did not express an intention to receive COVID-19 vaccination (mean VCBS = 32.2, SD = 11.2), compared to mean VCBS of 25.4 (SD = 9.2) and a mean VCBS of only 16.5 (SD = 8.0) among those who expressed an intention to receive COVID-19 vaccination (*p* < 0.001; ANOVA, Figure 4).

Univariate analysis showed that a higher VCBS score was correlated with COVID-19 vaccine hesitancy (yes vs. no and maybe; *p* < 0.001), with the following factors as covariates: age; sex; nationality; occupation; educational level; sector.

### 3.6. The 5C Psychological Determinants for COVID-19 Vaccination

To estimate the effect of each determinant of the 5C scale on the intention to receive COVID-19 vaccination, we evaluated the differences between the three groups (accepted, undecided, rejected). Higher levels of confidence and collective responsibility and lower levels of complacency, constraints, and calculation together with lower levels of vaccine conspiracy beliefs were associated with significant acceptance of COVID-19 vaccines, compared to hesitancy (maybe and no; Figure 5). Complete rejection for COVID-19 vaccination, compared to being unsure (maybe) was significantly associated with lower levels of confidence and collective responsibility, and higher levels of constraints and vaccine conspiracy beliefs (Figure 5). Complacency and calculation were not significantly linked with differences in vaccine acceptance among the hesitancy group (no vs. maybe; Figure 5). 

Multinomial logistic regression revealed that four of the 5C psychological determinants were significantly correlated with COVID-19 vaccine acceptance vs. COVID-19 vaccine hesitancy (no or maybe), with the following as covariates: age < 34 years vs. ≥34 years (*p* = 0.157); sex (*p* = 0.001); nationality (*p* = 0.284); occupation (*p* = 0.690); educational level (*p* = 0.349); workplace (*p* < 0.001; Figure 6). Lower levels of constraints and calculation were linked with COVID-19 vaccine acceptance (odds ratios: 8.1 and 2.9, respectively; *p* < 0.001 for both). Additionally, higher levels of confidence and collective responsibility were linked with COVID-19 vaccine acceptance (odds ratios: 2.3 (*p* = 0.001) and 5.7 (*p* < 0.001), respectively; Figure 6).

### 3.7. ROC Analysis of the 5C Subscales and the VCBS Scale

The evaluate the ability of VCBS and the five subscales of 5C in the prediction of COVID-19 vaccine acceptance, ROC analysis showed that the highest AUC was observed for constraints subscale (AUC = 0.872) followed by collective responsibility subscale (AUC = 0.843), confidence subscale (AUC = 0.839), VCBS (AUC = 0.820), complacency subscale (AUC = 0.807), while the lowest AUC was observed for the calculation subscale (AUC = 0.661; Figure 7a,b).

### 3.8. Sources of Knowledge about COVID-19 Vaccines and Its Relation to Vaccine Acceptance

Scientists and scientific journals were reported as the most common main source of knowledge regarding COVID-19 vaccines (*n* = 407, 39.9%), followed by doctors and other HCWs (*n* = 339, 33.3%). Dependence on social media platforms was reported at a higher rate among those who rejected COVID-19 vaccination, compared to those who expressed an intent to receive vaccination (23.9% vs. 17.0%); however, the differences lacked statistical significance (*p* = 0.085; χ^2^ test).

The dependence on social media platforms and on TV programs, newspapers, and news releases was associated with a significantly higher VCBS, compared to the dependence on scientists/scientific journals and doctors/other healthcare workers (*p* < 0.001; ANOVA, Figure 8). 

## 4. Discussion

The novelty in this study stems from being the first to assess the psychological antecedents for COVID-19 vaccination among HCWs in the Middle East region to the best of our knowledge. The focus on HCWs’ knowledge and attitudes toward COVID-19 vaccination is important for two reasons: First, HCWs represent a group with a higher risk for SARS-CoV-2 acquisition [44]. Second, HCWs can play a central role in addressing COVID-19 vaccine hesitancy by educating the patients, in particular, and the general public, in general, on the importance of vaccination in the fight against the ongoing pandemic. It was also important to assess the attitude of HCWs in Kuwait regarding COVID-19 vaccination considering the previous research in the Middle East region, and Kuwait, which has shown that COVID-19 vaccine acceptance rates were among the lowest globally [27,37,45,46].

The results of this study indicated an overall high intent to receive COVID-19 vaccines (83.3%) among HCWs in Kuwait. This rate was found to be in stark contrast with the previous reports from Kuwait and the Arab countries of the Middle East that reported the lowest rates of COVID-19 vaccine acceptance worldwide [27,37,43,47]. A plausible explanation of this difference might be related to the timing of the current survey (March 2021), during which several reports started to accrue that displayed the efficacy and safety of COVID-19 vaccination at a large level [6,12]. Additionally, Kuwait witnessed a surge in cases one month before the survey was distributed, with concurrent encouragement of HCWs to receive the vaccines.

Earlier studies evaluating the same objective reported wide variability in COVID-19 vaccine acceptance rates among HCWs [3,37,48,49]. The earlier reports that dated back to 2020 showed a higher level of vaccine hesitancy, compared to the recent reports [37,48,50]. This might be related to the growing evidence of the efficacy and safety of the currently used vaccines and their swift effect on the pandemic control, as shown in countries with high vaccine coverage [6,51]. The conspicuous differences in COVID-19 vaccine acceptance between the general public and HCWs were reported in a recent study from Poland, which showed an acceptance rate of 83.0% among HCWs, as opposed to 54.3% among the control group [52].

To put our result in a broader context, a recent study from Canada showed a close acceptance rate of 80.9% with males and physicians having a higher likelihood of vaccine acceptance [53]. A slightly lower COVID-19 vaccine acceptance rate of 73.1% was reported among HCWs in France, with a similar pattern of more medical staff in favor of vaccination, compared to nurses, while a closer rate of (78.5%) was reported among the Greek health professionals [54,55]. A recent study from Germany among emergency medical services personnel reported a lower acceptance rate of 57.0%, with males and participants of a higher educational level showing a higher propensity to vaccinate, which is in line with our findings [56]. A lower COVID-19 vaccine acceptance rate was also reported among Belgian hospital staff (58.0%) [57]. A much lower COVID-19 vaccine acceptance rate, of merely 21.0%, was reported among Egyptian HCWs [47]. In Saudi Arabia, the results showed an acceptance rate of 50.5%, with concerns about safety reported as a contributing factor to vaccine hesitancy in the study by Qattan et al. [58].

Further analysis revealed that certain variables were associated with a lower intent to receive COVID-19 vaccines among HCWs in Kuwait. These variables included female sex, non-Kuwaiti citizenship, being a nurse, or affiliation to the private sector, besides a lower educational level. A higher level of vaccine hesitancy among females appears as a recurrent pattern that was reported in various studies across different time periods and geographic locations [22,27,37,43,48,59,60]. COVID-19 vaccine hesitancy was also linked to lower educational levels in previous studies, and such a result appears as an expected outcome, considering the importance of knowledge as a driver for vaccine acceptance [27]. The higher level of vaccine hesitancy among participants affiliated with the private section might be related to possible constraints since vaccination campaigns are conducted and organized through the public health sector at the state level.

An important and recurring pattern seen in different studies is the higher prevalence of COVID-19 vaccine hesitancy among nurses [3,61]. The higher level of COVID-19 vaccine acceptance among physicians and dentists, compared to other HCWs, was reported previously (e.g., in Poland) and should be addressed with proper interventional measures including improving the trust levels in safety and efficacy of COVID-19 vaccines among nurses, who are considered among the highest risk groups for infection risk [3,57]. 

An interesting result of the current study was the finding that the majority of the study participants displayed a preference for mRNA vaccines (particularly for the Pfizer-BioNTech vaccine), followed by the preference for the Oxford-AstraZeneca vaccine. Specifically, more than two-thirds of the study participants showed a preference for the Pfizer-BioNTech vaccine. This might be related to the wide range of reports that have shown the superior efficacy of about 95%, compared to other currently available COVID-19 vaccines [6,14,62]. Another explanation can be related to the availability of this vaccine in Kuwait together with the Oxford-AstraZeneca vaccine. An additional result showed the declining preference toward the Oxford-AstraZeneca vaccine, which might be related to the controversy that erupted following the reporting of thromboembolic events with a potential link to the reception of such vaccine in a few European countries [63,64].

COVID-19 vaccine hesitancy in this study was associated with less preference toward mRNA vaccines. This can be explained by the recent use of this technology (used for the first time to prevent infectious disease in humans), and the potential fear of long-term side effects of such an innovative vaccine [65]. The concerns surrounding mRNA vaccines among the vaccine-hesitant groups can be tackled by providing sufficient reliable information about the excellent efficacy of such vaccines besides the emphasis on the theoretical evidence that points to the absence of long-term adverse effects [65,66].

An essential objective in this study was to assess the potential psychological factors that can contribute to COVID-19 vaccine hesitancy among HCWs in Kuwait. The 5C scale appears to have a satisfactory discrimination power in the prediction of the psychological determinants of COVID-19 vaccine acceptance [67]. Additionally, the use of the vaccine conspiracy beliefs scale has also been shown as an important tool in disclosing the role of conspiratorial beliefs in COVID-19 vaccine hesitancy [27,43]. For the psychological determinants for COVID-19 vaccine hesitancy among HCWs in Kuwait, the following patterns were found: four out of the 5Cs were significantly linked to a higher likelihood of accepting COVID-19 vaccination. Complacency was the only factor that did not show a statistical difference between acceptance and hesitancy groups. This might be related to the awareness and perception of COVID-19 dangers among HCWs [68,69].

For the other factors, previous studies have shown the link between vaccine acceptance and higher confidence levels, and the need for sufficient and accurate information about the vaccines among HCWs in Egypt, similar to our results [47]. Building trust in governments, policymakers, and pharmaceutical companies can be a decisive factor in increasing the levels of confidence in COVID-19 vaccination benefits, which, in turn, can increase the likelihood of recommending vaccination by HCWs [70,71,72].

In this study, constraints appeared as a major psychological factor that separates the vaccine acceptance group from the hesitancy group. This result points to the importance of identifying the physical and psychological barriers that would render COVID-19 vaccination an inconvenient experience. Such barriers can be reduced by strategies involving increasing the number of trained healthcare professionals capable of administering COVID-19 vaccines and launching newer vaccination centers.

Moreover, collective responsibility was observed as an important factor in COVID-19 vaccine acceptance in this study. The emphasis that individual vaccination would help to reach herd immunity necessary to control the ongoing pandemic should be a priority, which would help in increasing the number of HCWs willing to receive COVID-19 vaccines. Furthermore, calculation was another psychological factor with the difference between accepting and being hesitant to COVID-19 vaccination in this study. Providing sufficient knowledge on the several aspects of COVID-19 vaccination including its efficacy and value in reducing severity and mortality from the disease would help in COVID-19 vaccine acceptance. Another aspect of knowledge that should be provided is the cumulative evidence showing the safety of the currently approved vaccines and their potential value in reaching immunity at the population level, which, in turn, would reduce the burden on healthcare systems. For all the aforementioned psychological factors, ROC analyses showed that the 5C subscales (except for calculation subscale) could have potential use in predicting COVID-19 vaccine acceptance.

An additional important result was the finding of a significant link between holding vaccine conspiracy beliefs and vaccine hesitancy. The embrace of conspiratorial beliefs was prevalent since the early days of the COVID-19 pandemic, and later on, this extended to involve COVID-19 vaccination [26,27,73]. This included hoaxes such as the use of vaccines to insert quantum-dot spy software into the vaccinated individuals for monitoring purposes and the claims that mRNA vaccines will result in sterility [25]. The correlation between the beliefs in vaccine conspiracy with COVID-19 vaccine hesitancy was evident in this study and should be used to highlight the need for proper dismissal of all unsubstantiated allegations about COVID-19 vaccination. 

Finally, our results indicated the value of reliance on reliable sources to gain knowledge on the COVID-19 vaccine (e.g., scientific journals, scientists), which was linked to significantly higher rates of COVID-19 vaccine acceptance. This pattern was seen previously in different study groups (e.g., the general public, students), which shows the importance of vigilant fact-checking, particularly in social media platforms [27,43]. 

### Strengths and Limitations

The major strength of the study was the incorporation of 5C and VCBS to assess the psychological determinants of vaccine acceptance. In addition, the representation of a majority of HCW categories facilitated the identification of certain gaps in COVID-19 vaccine acceptance (higher hesitancy among nurses and laboratory workers).

Limitations of this study include the use of a convenient chain-referral sampling approach which might affect the generalizability of our results. Additionally, the internal consistency of the 5C subscales could have been improved by the addition of further items, particularly for the calculation subscale. Moreover, the cross-sectional nature of this study should not be overlooked since vaccine hesitancy is context specific, particularly for the place and time during which any survey is conducted.

Furthermore, the descriptive nature of this research can point to mere associations of the study variables. Hence, further studies with a more robust experimental design are needed before drawing definitive conclusions regarding the psychological determinants of COVID-19 vaccine hesitancy.

Finally, the substantial dependence on significance testing comes with obvious drawbacks that should be considered in future research [74,75]. 

## 5. Conclusions

An overall satisfactory level of COVID-19 vaccine acceptance was found among HCWs in Kuwait, which was found to be among the highest globally. However, certain gaps were identified including a higher rate of vaccine hesitancy among female HCWs, nurses, and HCWs in the private sector. Increasing the levels of trust and collective responsibility and reducing the possible physical and psychological constraints can be valuable to tackle the problem of COVID-19 vaccine hesitancy. Additionally, the proper dismissal of COVID-19 vaccination conspiratorial claims would also be helpful in addressing COVID-19 vaccine hesitancy. This can be achieved by reinforcing the importance of spreading clear messages through reliable sources (e.g., scientists and scientific journals), with emphasis on fact-checking of the messages conveyed through TV, newspapers, and social media platforms.

The high level of COVID-19 vaccine acceptance observed in this study can provide an important clue to the potential role that HCWs can play in educating their patients and the general public about the benefits of vaccination to limit the spread and severity of COVID-19. However, further studies with improved experimental design are needed to confirm such an association.

## Figures and Tables

**Figure 1 vaccines-09-00701-f001:**
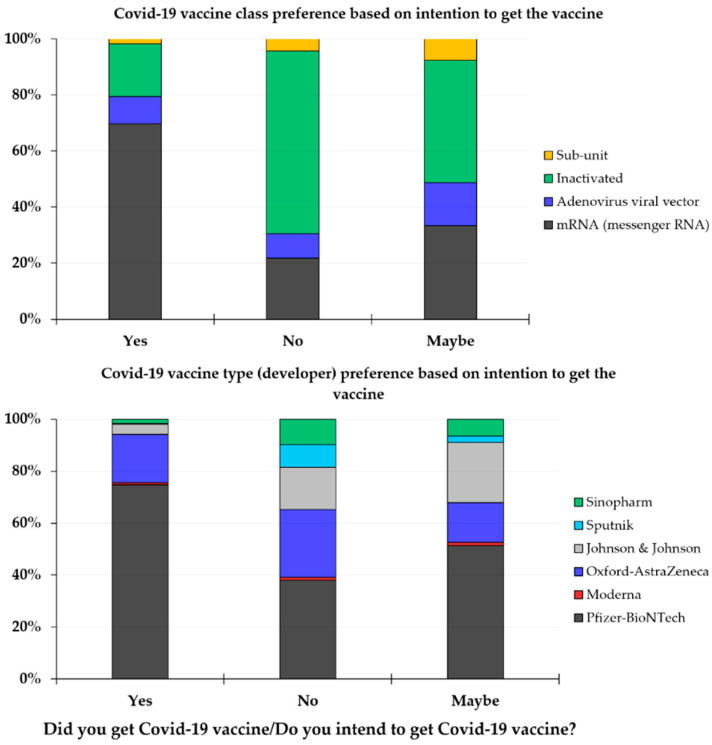
The preference of COVID-19 vaccine class and type (developer) stratified based on intention to receive vaccination.

**Figure 2 vaccines-09-00701-f002:**
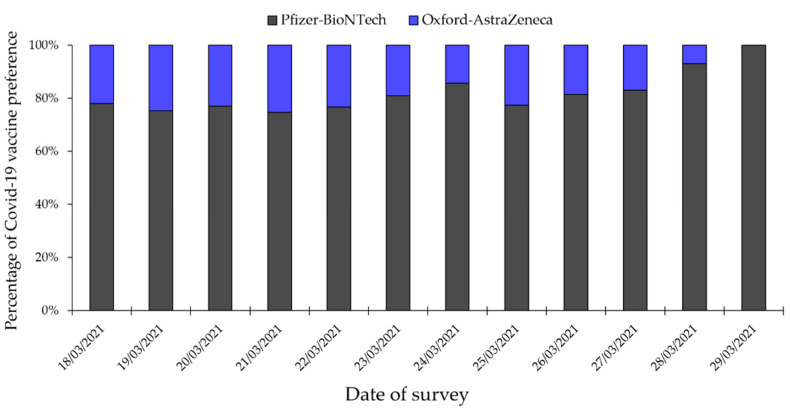
Change in COVID-19 vaccine preference over the survey dates (Pfizer-BioNTech vaccine vs. Oxford-AstraZeneca vaccine).

**Figure 3 vaccines-09-00701-f003:**
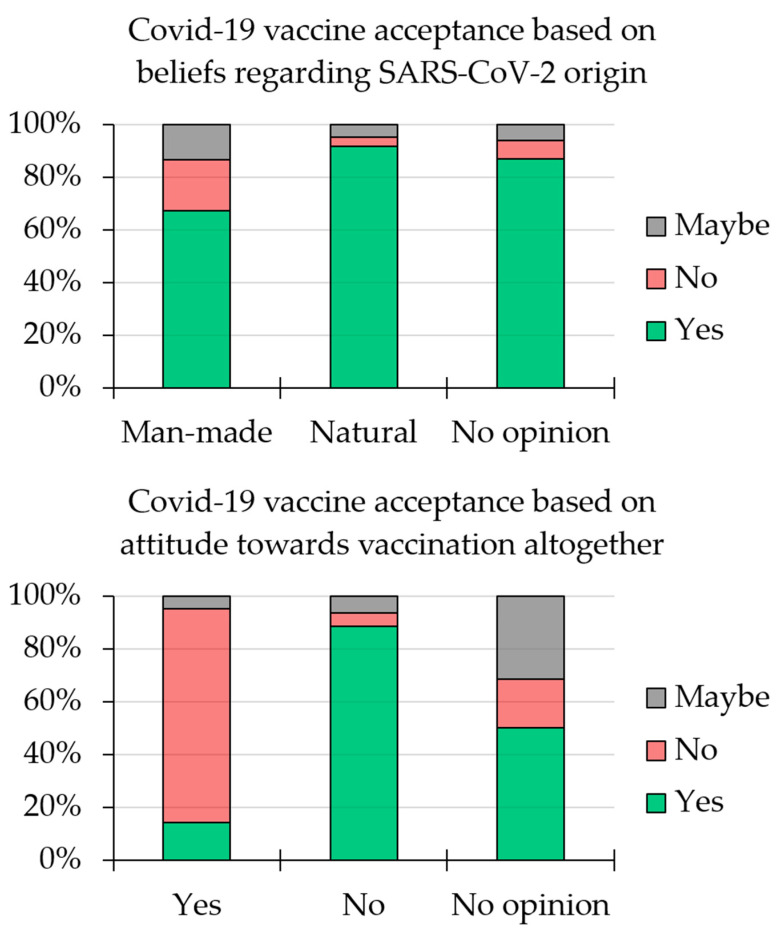
Intent to receive COVID-19 vaccination based on belief regarding virus origin and general attitude toward vaccination.

**Figure 4 vaccines-09-00701-f004:**
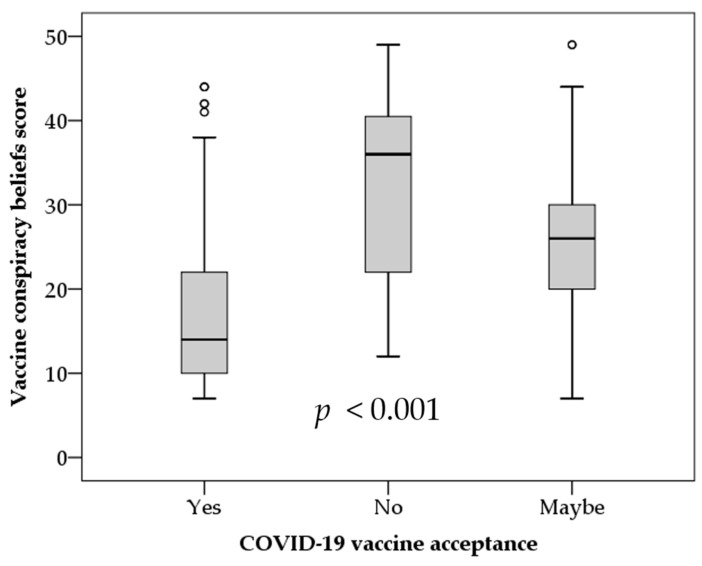
The correlation between vaccine conspiracy belief scale and intention to receive COVID-19 vaccination. *p* value was calculated using ANOVA test. The circles represent mild outliers.

**Figure 5 vaccines-09-00701-f005:**
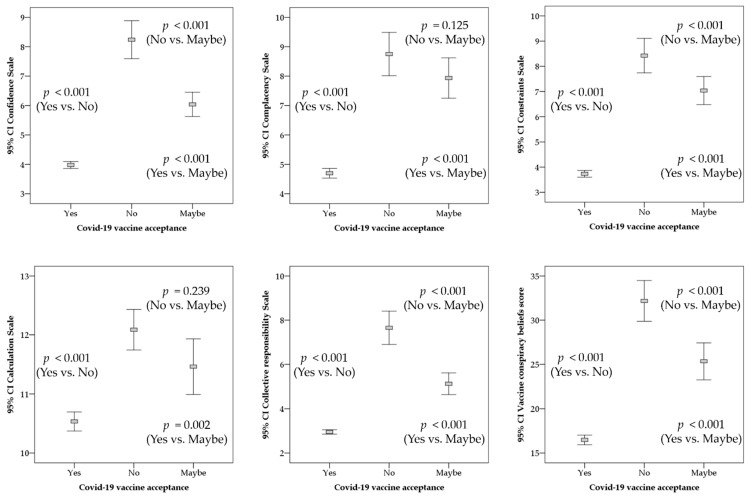
Analysis of COVID-19 vaccine acceptance among HCWs in Kuwait stratified based on the 5C subscales and the vaccine conspiracy belief scale (VCBS). *p* values are calculated based on ANOVA with Bonferroni corrections. CI: confidence interval of the mean.

**Figure 6 vaccines-09-00701-f006:**
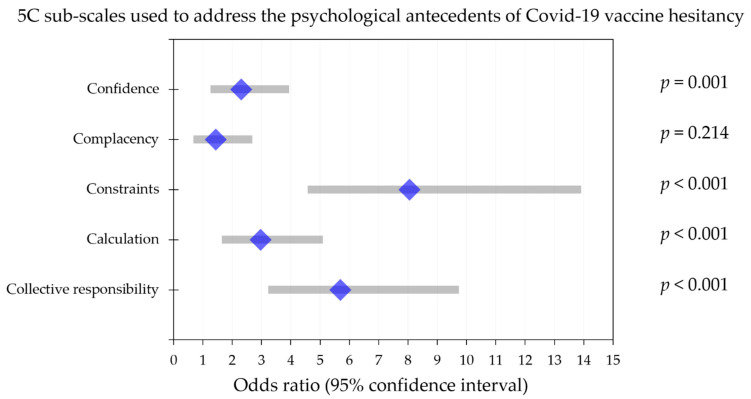
Multinomial regression analysis of the five psychological antecedents (subscales) for COVID-19 vaccine hesitancy and their association with COVID-19 vaccine hesitancy (intent to receive COVID-19 vaccination with no/maybe responses). The mean odds ratio is represented by the diamond blue shape, while the 95% confidence interval is displayed as the grey bar.

**Figure 7 vaccines-09-00701-f007:**
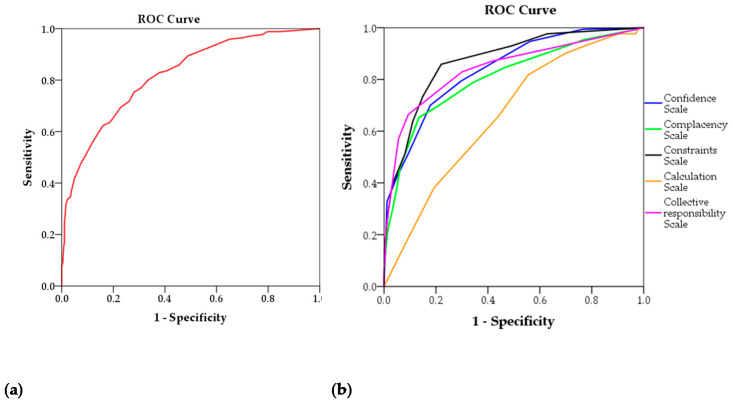
Receiver operating characteristic (ROC) analysis: (**a**) ROC curves of the VCBS scale in COVID-19 vaccine acceptance screening; (**b**) ROC curves for 5C subscales in COVID-19 vaccine acceptance screening.

**Figure 8 vaccines-09-00701-f008:**
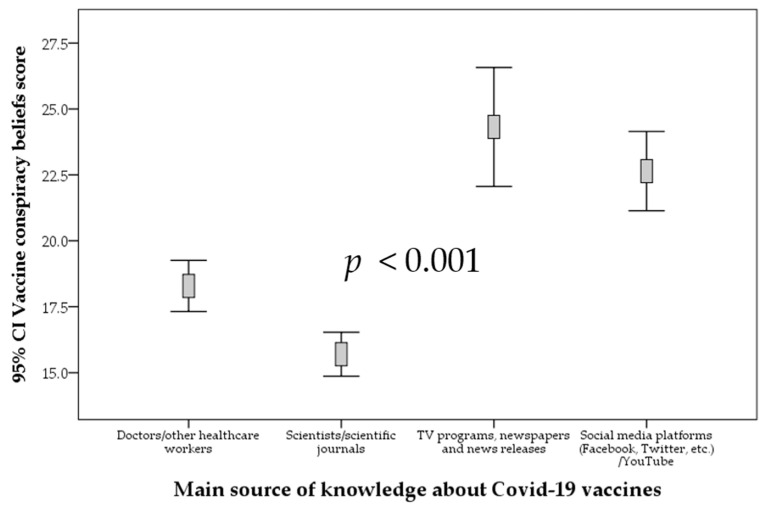
The correlation between vaccine conspiracy belief scale and the main source of knowledge about COVID-19 vaccines. *p* value was calculated using ANOVA test. CI: confidence interval of the mean.

**Table 1 vaccines-09-00701-t001:** The overall characteristics of the study participants.

Characteristic	Feature	N ^3^ (%)
Mean age (SD ^1^)		33.6 (9.7)
Sex	Male	393 (38.6)
	Female	626 (61.4)
Nationality	Kuwaiti	765 (75.1)
	Non-Kuwaiti	221 (21.7)
	Stateless/unknown	33 (3.2)
Occupation	Physician	292 (28.7)
	Dentist	170 (16.7)
	Pharmacist	206 (20.2)
	Nurse	127 (12.5)
	Laboratory technician	80 (7.9)
	Other ^2^	144 (14.1)
Educational level	Undergraduate degree	662 (65.0)
	Postgraduate degree	357 (35.0)
Workplace	Public	915 (89.8)
	Private	104 (10.2)
History of chronic disease	Yes	182 (17.9)
	No	837 (82.1)
Experience of COVID-19 in self or family	Yes	523 (51.3)
	No	496 (48.7)

^1^ SD: standard deviation; ^2^ Other: other categories of healthcare workers (e.g., physiotherapists; dieticians and nutritionists; optometrists, etc.); ^3^ N: number.

**Table 2 vaccines-09-00701-t002:** The overall characteristics of the study participants stratified by occupation.

Characteristic	Feature	Occupation N ^2^ (%)						*p* Value ^4^
		Physician	Dentist	Pharmacist	Nurse	Laboratory Technician	Other ^3^	
Age	Mean (SD ^1^)	36 (11.0)	31 (7.1)	34 (8.4)	33 (9.2)	35 (10.7)	32 (10.1)	0.001
Sex	Male	150 (51.4)	101 (59.4)	45 (21.8)	33 (26.0)	25 (31.3)	39 (27.1)	<0.001
	Female	142 (48.6)	69 (40.6)	161 (78.2)	94 (74.0)	55 (68.8)	105 (72.9)	
Nationality	Kuwaiti	229 (78.4)	148 (87.1)	188 (91.3)	33 (26.0)	51 (63.7)	116 (80.6)	<0.001
	Non-Kuwaiti	56 (19.2)	17 (10.0)	16 (7.8)	88 (69.3)	22 (27.5)	22 (15.3)	
	Stateless/unknown	7 (2.4)	5 (2.9)	2 (1.0)	6 (4.7)	7 (8.8)	6 (4.2)	
Educational level	Undergraduate degree	116 (39.7)	105 (61.8)	141 (68.4)	120 (94.5)	66 (82.5)	114 (79.2)	<0.001
	Postgraduate degree	176 (60.3)	65 (38.2)	65 (31.6)	7 (5.5)	14 (17.5)	30 (20.8)	
Workplace	Public	276 (94.5)	152 (89.4)	175 (85.0)	112 (88.2)	77 (96.3)	123 (85.4)	0.002
	Private	16 (5.5)	18 (10.6)	31 (15.0)	15 (11.8)	3 (3.8)	21 (14.6)	

^1^ SD: standard deviation; ^2^ N: number; ^3^ Other: other categories of healthcare workers (e.g., physiotherapists; dieticians and nutritionists; optometrists, etc.); ^4^ *p* value: calculated using chi-squared test except for age which was calculated using the Kruskal–Wallis test.

**Table 3 vaccines-09-00701-t003:** Association of participants’ characteristics with intent to receive COVID-19 vaccination.

Variable	Feature	Did You Get COVID-19 Vaccine/Do You Intend to Get COVID-19 Vaccine? N ^3^ (%)			*p* Value ^4^
		Yes	No	Maybe	
Mean age (SD ^1^)		33.9 (9.9)	31.8 (8.7)	32.2 (8.1)	0.208
Sex	Male	359 (91.3)	15 (3.8)	19 (4.8)	<0.001 ***
	Female	490 (78.3)	77 (12.3)	59 (9.4)	
Nationality	Kuwaiti	651 (85.1)	65 (8.5)	49 (6.4)	0.031 *
	Non-Kuwaiti	172 (77.8)	22 (10.0)	27 (12.2)	
	Stateless/unknown	26 (78.8)	5 (15.2)	2 (6.1)	
Occupation	Physician	264 (90.4)	13 (4.5)	15 (5.1)	<0.001 ***
	Dentist	155 (91.2)	5 (2.9)	10 (5.9)	
	Pharmacist	165 (80.1)	26 (12.6)	15 (7.3)	
	Nurse	89 (70.1)	22 (17.3)	16 (12.6)	
	Laboratory technician	61 (76.3)	11 (13.8)	8 (10.0)	
	Other ^2^	115 (79.9)	15 (10.4)	14 (9.7)	
Educational level	Undergraduate degree	530 (80.1)	71 (10.7)	61 (9.2)	0.001 **
	Postgraduate degree	319 (89.4)	21 (5.9)	17 (4.8)	
Workplace	Public	773 (84.5)	80 (8.7)	62 (6.8)	0.004 **
	Private	76 (73.1)	12 (11.5)	16 (15.4)	
History of chronic disease	Yes	147 (80.8)	24 (13.2)	11 (6.0)	0.075
	No	702 (83.9)	68 (8.1)	67 (8.0)	
Experience of COVID-19 in self or family	Yes	434 (83.0)	51 (9.8)	38 (7.3)	0.654
	No	415 (83.7)	41 (8.3)	40 (8.1)	

^1^ SD: standard deviation; ^2^ Other: other categories of healthcare workers (e.g., physiotherapists; dieticians and nutritionists; optometrists, etc.); ^3^ N: number; ^4^ *p* value: calculated using chi-squared test except for age which was calculated using the Kruskal–Wallis test; one asterisk indicates significant results <0.050 and ≥0.010; two asterisks indicate significant results <0.010 and ≥0.001; three asterisks indicate significant results <0.001.

**Table 4 vaccines-09-00701-t004:** Association of vaccine class preference with participants’ characteristics.

Variable	Feature	COVID-19 Vaccine Class N ^3^ (%)				*p* Value ^4^
		mRNA ^2^	Viral Vector	Sub-unit	Inactivated	
Sex	Male	278 (70.7)	31 (7.9)	3 (0.8)	81 (20.6)	<0.001 ***
	Female	360 (57.5)	72 (11.5)	22 (3.5)	172 (27.5)	
Nationality	Kuwaiti	486 (63.5)	83 (10.8)	21 (2.7)	175 (22.9)	0.084
	Non-Kuwaiti	129 (58.4)	20 (9.0)	4 (1.8)	68 (30.8)	
	Stateless/unknown	23 (69.7)	0	0	10 (30.3)	
Occupation	Physician	216 (74.0)	17 (5.8)	2 (0.7)	57 (19.5)	<0.001 ***
	Dentist	125 (73.5)	13 (7.6)	1 (0.6)	31 (18.2)	
	Pharmacist	120 (58.3)	40 (19.4)	3 (1.5)	43 (20.9)	
	Nurse	57 (44.9)	13 (10.2)	6 (4.7)	51 (40.2)	
	Laboratory technician	43 (53.8)	6 (7.5)	4 (5.0)	27 (33.8)	
	Other ^1^	77 (53.5)	14 (9.7)	9 (6.3)	44 (30.6)	
Educational level	Undergraduate degree	399 (60.3)	72 (10.9)	19 (2.9)	172 (26.0)	0.164
	Postgraduate degree	239 (66.9)	31 (8.7)	6 (1.7)	81 (22.7)	
Workplace	Public	575 (62.8)	94 (10.3)	22 (2.4)	224 (24.5)	0.839
	Private	63 (60.6)	9 (8.7)	3 (2.9)	29 (27.9)	
History of chronic disease	Yes	118 (64.8)	18 (9.9)	4 (2.2)	42 (23.1)	0.915
	No	520 (62.1)	85 (10.2)	21 (2.5)	211 (25.2)	
Experience of COVID-19 in self or family	Yes	340 (65.0)	52 (9.9)	11 (2.1)	120 (22.9)	0.378
	No	298 (60.1)	51 (10.3)	14 (2.8)	133 (26.8)	

^1^ Other: other categories of healthcare workers (e.g., physiotherapists; dieticians and nutritionists; optometrists, etc.); ^2^ mRNA: messenger RNA; ^3^ N: number; ^4^ *p* value: calculated using chi-squared test; three asterisks indicate significant results <0.001.

**Table 5 vaccines-09-00701-t005:** Association of vaccine type (developer) preference with participants’ characteristics.

Variable	Feature	COVID-19 Vaccine Type (Developer) N ^2^ (%)						*p* Value ^3^
		Pfizer-BioNTech	Sinopharm	Oxford-AstraZeneca	Sputnik	Moderna	Johnson & Johnson	
Sex	Male	298 (75.8)	11 (2.8)	58 (14.8)	4 (1.0)	2 (0.5)	20 (5.1)	0.027 *
	Female	412 (65.8)	17 (2.7)	135 (21.6)	9 (1.4)	7 (1.1)	46 (7.3)	
Nationality	Kuwaiti	544 (71.1)	14 (1.8)	141 (18.4)	11 (1.4)	7 (0.9)	48 (6.3)	0.024 *
	Non-Kuwaiti	138 (62.4)	14 (6.3)	48 (21.7)	2 (0.9)	2 (0.9)	17 (7.7)	
	Stateless/unknown	28 (84.8)	0	4 (12.1)	0	0	1 (3.0)	
Occupation	Physician	224 (76.7)	7 (2.4)	43 (14.7)	3 (1.0)	3 (1.0)	12 (4.1)	<0.001 ***
	Dentist	144 (84.7)	2 (1.2)	16 (9.4)	0	1 (0.6)	7 (4.1)	
	Pharmacist	125 (60.7)	4 (1.9)	54 (26.2)	1 (0.5)	0	22 (10.7)	
	Nurse	72 (56.7)	5 (3.9)	38 (29.9)	2 (1.6)	1 (0.8)	9 (7.1)	
	Laboratory technician	52 (65.0)	3 (3.8)	14 (17.5)	3 (3.8)	1 (1.3)	7 (8.8)	
	Other ^1^	93 (64.6)	7 (4.9)	28 (19.4)	4 (2.8)	3 (2.1)	9 (6.3)	
Educational level	Undergraduate degree	443 (66.9)	20 (3.0)	139 (21.0)	9 (1.4)	6 (0.9)	45 (6.8)	0.205
	Postgraduate degree	267 (74.8)	8 (2.2)	54 (15.1)	4 (1.1)	3 (0.8)	21 (5.9)	
Workplace	Public	645 (70.5)	22 (2.4)	173 (18.9)	13 (1.4)	9 (1.0)	53 (5.8)	0.018 *
	Private	65 (62.5)	6 (5.8)	20 (19.2)	0	0	13 (12.5)	
History of chronic disease	Yes	120 (65.9)	7 (3.8)	41 (22.5)	3 (1.6)	3 (1.6)	8 (4.4)	0.295
	No	590 (70.5)	21 (2.5)	152 (18.2)	10 (1.2)	6 (0.7)	58 (6.9)	
Experience of COVID-19 in self or family	Yes	364 (69.6)	13 (2.5)	97 (18.5)	3 (0.6)	4 (0.8)	42 (8.0)	0.122
	No	346 (69.8)	15 (3.0)	96 (19.4)	10 (2.0)	5 (1.0)	24 (4.8)	

^1^ Other: other categories of healthcare workers (e.g., physiotherapists; dieticians and nutritionists; optometrists, etc.); ^2^ N: Number; ^3^ *p* value: calculated using chi-squared test; one asterisk indicates significant results <0.050 and ≥0.010; three asterisks indicate significant results <0.001.

## Data Availability

The raw data collected in this study are available on request from the corresponding author (M.S.).

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
