# Peer review of "Psychological Determinants of COVID-19 Vaccine Acceptance among Healthcare Workers in Kuwait: A Cross-Sectional Study Using the 5C and Vaccine Conspiracy Beliefs Scales"

_vaccines, 2021, doi:10.3390/vaccines9070701_

Round 1

Reviewer 1 Report

The manuscript is well written and provide information useful to further understand hesitancy among HCW in Kuwait. I was not able to link to the supplemental material via the suggested url www.mdpi.com/xxx/s1 It is very important that the reader has direct access to the survey questions. 

Author Response

Reviewer #1 comments

The manuscript is well written and provide information useful to further understand hesitancy among HCW in Kuwait. I was not able to link to the supplemental material via the suggested url www.mdpi.com/xxx/s1 It is very important that the reader has direct access to the survey questions.

Response: We would like to thank the reviewer for the comments. Regarding the access to the Supplementary File, we will ensure that it was uploaded successfully. In our initial submission we provided this supplementary file entitled “vaccines-1265766-supplementary.docx”, as a word document and we will re-upload it in our revision.

Reviewer 2 Report

The piece is reasonably understandable, but I have the following issues.

  1. The most important issue is that the manuscript is descriptive. Thus, there is no way to come to valid causal conclusions. For example, the authors state: “Increasing the levels of trust and collective responsibility 533 and reducing the possible physical and psychological constraints would be valuable to 534 tackle the problem of Covid-19 vaccine hesitancy. Additionally, the proper dismissal of 535 Covid-19 vaccination conspiratorial claims would also be helpful in addressing Covid-19 536 vaccine hesitancy. This can be achieved by re-enforcing the importance of spreading clear 537 messages through reliable sources (e.g. scientists and scientific journals), with emphasis 538 on fact-checking of the messages conveyed through TV, newspapers and social media 539 platforms.” However, I think these conclusions are not justified as all the author has are associations. It would take an additional study, with a much stronger experimental paradigm, to justify the conclusions.

Moreover, the last sentence says: “The high level of Covid-19 vaccine acceptance observed in this study can provide an 541 important clue to the potential role that HCWs can play in educating their patients and 542 the general public to the benefits of vaccination to limit the spread and severity of Covid- 543 19.” However, I again think the descriptive nature of the research is far too premature to come to conclusions like this. The truth is that until someone attempts to manipulate the alleged crucial variables, we will not know whether there is anything important here.

  1. I think there is way too much dependence on significance testing. At the risk of being self-serving, the author might consult one of my papers in Econometrics, which is a MDPI journal.

Trafimow, D. (2019). A frequentist alternative to significance testing, p-values, and confidence intervals. Econometrics, 7(2), 1-14. https://www.mdpi.com/2225-1146/7/2/26

  1. There is also the issue of the sampling. The assumption underlying the statistics used is that there was random and independent sampling. However, the authors admit to having used “convenience” sampling. Therefore, in addition to this being a statistical problem, it is also a generalization problem. What assurance do we have, for example, that the sample statistics accurately estimate the corresponding population parameters?

In summary, I am not sure what to recommend. On the positive side, the manuscript suggests variables that are of potential importance to manipulate and this might be enough reason to publish. On the other hand, there is the inability to validly determine causal relationships, what I consider too much dependence on significance testing, and a lack of representative sampling from the population.

I sign my reviews, David Trafimow

Author Response

Reviewer #2 comments (Dr. David Trafimow)

The piece is reasonably understandable, but I have the following issues.

The most important issue is that the manuscript is descriptive. Thus, there is no way to come to valid causal conclusions. For example, the authors state: “Increasing the levels of trust and collective responsibility 533 and reducing the possible physical and psychological constraints would be valuable to 534 tackle the problem of Covid-19 vaccine hesitancy. Additionally, the proper dismissal of 535 Covid-19 vaccination conspiratorial claims would also be helpful in addressing Covid-19 536 vaccine hesitancy. This can be achieved by re-enforcing the importance of spreading clear 537 messages through reliable sources (e.g. scientists and scientific journals), with emphasis 538 on fact-checking of the messages conveyed through TV, newspapers and social media 539 platforms.” However, I think these conclusions are not justified as all the author has are associations. It would take an additional study, with a much stronger experimental paradigm, to justify the conclusions.

Response: We are deeply grateful for this insightful comment. We agree with the reviewer’s comment regarding this issue and based on this, we added the following paragraph to the limitations section: “Furthermore, the descriptive nature of this research can point to mere associations of the study variables. Hence, further studies with a more robust experimental design are needed before drawing definitive conclusions regarding the psychological determinants of Covid-19 vaccine hesitancy.”

Moreover, the last sentence says: “The high level of Covid-19 vaccine acceptance observed in this study can provide an 541 important clue to the potential role that HCWs can play in educating their patients and 542 the general public to the benefits of vaccination to limit the spread and severity of Covid- 543 19.” However, I again think the descriptive nature of the research is far too premature to come to conclusions like this. The truth is that until someone attempts to manipulate the alleged crucial variables, we will not know whether there is anything important here.

Response: Again, we are deeply grateful for this meticulous comment. We totally agree with the reviewer’s comment and based on this; we added the following paragraph to the conclusions section: “However; further studies with improved experimental design are needed to confirm such an association.”

I think there is way too much dependence on significance testing. At the risk of being self-serving, the author might consult one of my papers in Econometrics, which is a MDPI journal.

Trafimow, D. (2019). A frequentist alternative to significance testing, p-values, and confidence intervals. Econometrics, 7(2), 1-14. https://www.mdpi.com/2225-1146/7/2/26

Response: We are thankful for this comment that enlightened us regarding potential pitfalls in this research which should have been previously mentioned in the limitations. Thus, we added the following paragraph to the limitations section: “Finally, the substantial dependence on significance testing comes with obvious drawbacks that should be considered in future research [77,78].”

Additionally, we added the following relevant references:

https://doi.org/10.3390/econometrics7020026

https://doi.org/10.1016/j.newideapsych.2017.01.002

There is also the issue of the sampling. The assumption underlying the statistics used is that there was random and independent sampling. However, the authors admit to having used “convenience” sampling. Therefore, in addition to this being a statistical problem, it is also a generalization problem. What assurance do we have, for example, that the sample statistics accurately estimate the corresponding population parameters?

In summary, I am not sure what to recommend. On the positive side, the manuscript suggests variables that are of potential importance to manipulate and this might be enough reason to publish. On the other hand, there is the inability to validly determine causal relationships, what I consider too much dependence on significance testing, and a lack of representative sampling from the population.

I sign my reviews, David Trafimow

Response: We are grateful for the insightful comments made by the reviewer (Dr. David Trafimow) and we hope that the previous additions to the limitations and conclusions section (addressing the issues of sampling and significance testing) would be sufficient to make our manuscript suitable for publication.

Thanks!

Reviewer 3 Report

This survey focused on the acceptance of COVID-19 vaccines of healthcare workers in Kuwait. This manuscript should be further strengthened by addressing a  concern as follow: 

Minor: 

Please check the interquartile range: 32-39 years in line 235 are correct because the median age is also 32 years.

Author Response

Reviewer #3 comment

This survey focused on the acceptance of COVID-19 vaccines of healthcare workers in Kuwait. This manuscript should be further strengthened by addressing a concern as follow:

Minor:

Please check the interquartile range: 32-39 years in line 235 are correct because the median age is also 32 years.  

Response: We would like to thank the reviewer for the comments and careful review of our manuscript. Based on the reviewer comment, and after referring to the original data file, we corrected the IQR into 26-39 (Page 5, line 235).

Reviewer 4 Report

The topic is of interest, but of limited scope.

Some recent articles:

R.M. Ghazy et al., Determining the Cutoff Points of the 5C Scale for
Assessment of COVID-19 Vaccines Psychological Antecedents among the Arab
Population: A Multinational Study. Journal of Primary Care and Community
Health 2021, 12.

Area et al., One year of the covid-19 pandemic in galicia: A global view
of age-group statistics during three waves. International Journal of
Environmental Research and Public Health 18 (2021), 5104.

The authors indicate that “The novelty in this study stems from being
the first to assess the psychological antecedents for Covid-19
vaccination among HCWs in the Middle East region to the best of our
knowledge”
but, you can find some published works even in this journal.

The phrase “The high level of Covid-19 vaccine acceptance observed in
this study can provide an important clue to the potential role that HCWs
can play in educating their patients and the general public to the
benefits of vaccination to limit the spread and severity of Covid-19.”
does not reveal the origin or causes of the results.

Author Response

Reviewer #4 comments

The topic is of interest, but of limited scope.

Some recent articles:

R.M. Ghazy et al., Determining the Cutoff Points of the 5C Scale for Assessment of COVID 19 Vaccines Psychological Antecedents among the Arab Population: A Multinational Study. Journal of Primary Care and Community Health 2021, 12.

Area et al., One year of the covid-19 pandemic in galicia: A global view of age-group statistics during three waves. International Journal of Environmental Research and Public Health 18 (2021), 5104.

The authors indicate that “The novelty in this study stems from being the first to assess the psychological antecedents for Covid-19 vaccination among HCWs in the Middle East region to the best of our knowledge” but, you can find some published works even in this journal.

Response: We would like to thank the reviewer for the comments; however, we disagree regarding the point that other published work tackled the same objective in our study (assessing the psychological determinants of Covid-19 vaccine acceptance among HCWs in Kuwait).

In conducting the literature review for this study, we were aware about the excellent publication by Dr. Ghazy et al, which was the first to use a validated 5C (in Arabic). We previously cited this paper in reference No. 69: Ghazy, R.M.; Abd ElHafeez, S.; Shaaban, R.; Elbarazi, I.; Abdou, M.S.; Ramadan, A.; Kheirallah, K.A. Determining the Cutoff Points of the 5C Scale for Assessment of COVID-19 Vaccines Psychological Antecedents among the Arab Population: A Multinational Study. Journal of Primary Care & Community Health 2021, 12: 21501327211018568, doi:10.1177/21501327211018568.

Thus, we prefer to keep the discussion in the current format.

The phrase “The high level of Covid-19 vaccine acceptance observed in this study can provide an important clue to the potential role that HCWs can play in educating their patients and the general public to the benefits of vaccination to limit the spread and severity of Covid-19.” does not reveal the origin or causes of the results.

Response: Regarding this statement, it was based on the previous accumulating evidence of low levels of Covid-19 vaccine acceptance in the Arab countries of the Middle East region. Considering the high rates of Covid-19 vaccine acceptance among HCWs in Kuwait who participated in the study, this can be used as a starting point to highlight the important role that HCWs can play (since a majority of this group are not hesitant about the importance of vaccination) in educating the public regarding the benefits of vaccination.

Thus, we prefer to keep the discussion in the current format.

Reviewer 5 Report

Estimated Authors,

I've read with interest the present paper from Al-Sanafi and Sallam.

The study deals with the Psychological Determinants of Covid-19 Vaccine Acceptance Among Healthcare Workers in Kuwait. Authors have applied the well-known and highly reliable 5C scale from Betsch et al, with minimal adjustments required by the settings of the study.

Introduction is properly defining the settings and the aims. Methods are clearly reported and results are easy to follow. Discussion is clearly and properly written. The text is free from significant English mistakes/typos.

In summary, in my opinion, this is a high quality paper that may be accepted as it is, an option that I endorse.

Author Response

Reviewer #5 comment

Estimated Authors,

I've read with interest the present paper from Al-Sanafi and Sallam.

The study deals with the Psychological Determinants of Covid-19 Vaccine Acceptance Among Healthcare Workers in Kuwait. Authors have applied the well-known and highly reliable 5C scale from Betsch et al, with minimal adjustments required by the settings of the study.

Introduction is properly defining the settings and the aims. Methods are clearly reported and results are easy to follow. Discussion is clearly and properly written. The text is free from significant English mistakes/typos.

In summary, in my opinion, this is a high quality paper that may be accepted as it is, an option that I endorse.  

Response: We are deeply grateful, and we would like to thank the reviewer for his/her comments.

We are deeply grateful for the comprehensive, insightful and thorough review of our manuscript.

Thank you!

Round 2

Reviewer 2 Report

The authors have addressed my issues. Therefore, I am willing to support publication. 

Reviewer 4 Report

No comments